# Low-Fluorine Ba-Deficient Solutions for High-Performance Superconducting YBCO Films

**Pau Ternero** [1], **Jordi Alcalà** [1], **Laura Piperno** [2], **Cornelia Pop** [1], **Susagna Ricart** [1], **Narcis Mestres** [1], **Xavier Obradors** [1], **Teresa Puig** [1], **Giovanni Sotgiu** [2], **Giuseppe Celentano** [3] and **Anna Palau** [1,*]

1   Institut de Ciència de Materials de Barcelona, ICMAB-CSIC, Campus UAB, 08193 Bellaterra, Spain;
    pau.ternero@gmail.com (P.T.); jalcala@icmab.es (J.A.); cpop@icmab.es (C.P.); ricart@icmab.es (S.R.);
    narcis.mestres@icmab.es (N.M.); xavier.obradors@icmab.es (X.O.); teresa.puig@icmab.es (T.P.)
2   Engineering Department, Roma Tre University, Via Vito Volterra, 62, 00146 Rome, Italy;
    laura.piperno@uniroma3.it (L.P.); giovanni.sotgiu@uniroma3.it (G.S.)
3   ENEA, Frascati Research Centre, Via E. Fermi, 45-00044 Frascati, Italy; giuseppe.celentano@enea.it
*   Correspondence: palau@icmab.es

**Abstract:** High-performing superconducting $YBa_2Cu_3O_{7-x}$ (YBCO) films are fabricated by a chemical solution deposition methodology through novel barium-deficient low-fluorine solutions. The precursor solutions, distinguished for being straightforward, inexpensive and eco-friendly, allow us to reduce the growing temperature of YBCO down to 750 °C. We investigated the influence of the growing temperatures on both the microstructure and superconducting properties of YBCO films by using conventional thermal annealing and flash-heating approaches. A clear correlation between the growing temperature ($T_g$) and the superconducting performance of the films was obtained with improved performances observed at low $T_g$.

**Keywords:** high-temperature superconductors; critical current density; vortex pinning; chemical solution deposition





## 1. Introduction

Cuprates, and particularly rare-earth cuprates, constitute a class of deeply investigated functional oxides since they were found to be high-temperature superconductors (HTSs) [1]. Among HTS cuprate materials, $YBa_2Cu_3O_{7-x}$ (YBCO) has attracted most of the attention of researchers because its superconducting features make it a good candidate for the fabrication of high current cables and power devices based on YBCO coated conductors [2,3].

The technologies to grow YBCO thin films can be divided into physical and chemical methods. On the one hand, vacuum methods comprise the most widely studied techniques like pulsed laser deposition (PLD) [4], physical vapor deposition (PVD) [5] and metal-organic chemical vapor deposition (MOCVD) [6,7]. They require ultra-high vacuum systems, which raise the production costs and cause difficulties in scaling up the length of the process. On the other hand, representative non-vacuum methods are chemical solution deposition (CSD) techniques, which are currently in an advanced development stage [8–10]. The recent progress made on CSD YBCO growth and the feasibility of adapting this approach to large scale production processes has led this process to be considered as a low-cost, reliable and long-lasting coated conductor fabrication technique. Furthermore, CSD offers additional advantages such as an accurate control of the stoichiometric composition, a high deposition speed even on large areas and a high homogeneity of the deposited layer, limiting the formation of non-superconducting phases. The most widely employed methodology among CSD growth routes of YBCO is metal-organic decomposition (MOD) [11], in which the organic matrix that surrounds the metal ions is removed by thermal decomposition.

Several metal-organic salts have been successfully used: acetylacetonates [12], acetates [13], propionates [14], and trifluoroacetates [15], etc. Among them, the most widespread procedure for YBCO deposition is the one involving trifluoroacetate (TFA) salts [16]. However, the TFA-MOD method has several disadvantages such as the extreme sensitivity to humidity in the air and the high concentration of fluorine in the starting solutions, which leads to a non-eco-friendly route [17]. Nevertheless, the presence of fluorine is essential for the growth of the superconductor if one wants to avoid alternative routes using fluorine-free precursors, which have $BaCO_3$ decomposition as an intermediate phase [16,18,19]. Thus, a reduction of the fluorine content to that strictly necessary to form $BaF_2$ as a sub-product from the pyrolysis was investigated [14,20,21].

The stoichiometric composition of the precursor solutions has been proven to have a remarkable influence on the final properties of the samples [22]. The ratio of the metal-organic salts of Y/Ba/Cu = 1:1.5:3 (Ba-deficient) has been shown to be effective for obtaining YBCO films with higher critical current density ($J_c$) than that obtained for YBCO layers fabricated with the standard ratio Y/Ba/Cu = 1:2:3 [23]. However, there is little knowledge on the mechanism of influence of the starting solution composition on the superconductive properties of YBCO films. A first hypothesis links the improvement of transport properties to a denser, more homogeneous microstructure with reduced porosity and segregation of secondary phases [24].

In this study, a Ba-deficient low-fluorine TFA route, characterized as straightforward, inexpensive, eco-friendly and high-performing, is presented. The superconducting performance of the resulting YBCO films as a function of the crystallization temperature was investigated in order to find the appropriate growth temperature ($T_g$).

## 2. Materials and Methods

The starting solutions were prepared by dissolving barium and copper acetates, and yttrium trifluoroacetate (low-F) with the ratio of Y/Ba/Cu equal to 1:1.5:3 (Ba-deficient) in a mixture of organic solvents consisting of propionic acid (30%) and methanol while heating at 30 °C. The choice of short chain carboxylates, such as acetates and trifluoroacetates, was due to their decomposition behavior, which has been widely studied [25]. The choice of the solvent was based on previous studies performed on YBCO samples grown with TFA solutions [8]. It is important to note, however, that having an alcohol and a carboxylic acid in the reaction mixture favors the formation of water by means of an esterification reaction, which may have a detrimental effect on its superconducting performance [26]. Therefore, the solvents were smoothly evaporated until constant weight (approximately 2.5 h) in a vacuum rotary evaporator at 131 mbar and with a bath temperature of 70 °C to ensure a water content less than 0.6%. The gelatinous textures obtained were dissolved in small amounts of methanol and homogenized by stirring for 24 h. Afterwards, the solutions were brought to the desired concentration (1.5 M) with additional methanol and filtered through 0.22 μm filters to avoid impurities and guarantee homogeneity in the final solutions.

These precursor solutions were deposited on $5 \times 5$ mm$^2$ (100) LaAlO$_3$ (LAO) single-crystal substrates by spin coating with an angular acceleration of 6000 rpm/s, an angular velocity of 6000 rpm and a spinning duration time of 2 min. After spinning, in order to favor the solvent evaporation, the substrate was positioned over a crucible situated on a hot-plate at 70 °C.

Two types of heat treatments were applied to these samples: low temperature calcination (pyrolysis) and successive high temperature annealing (growth) to crystallize the YBCO phase, followed by a final oxygenation step. The pyrolysis processes were performed with a constant O$_2$ flow of 120 mL/min. The samples were heated up to 240 °C with a heating rate of 5 °C/min. The atmosphere was humidified to avoid copper salt sublimation. This was performed when the temperature was above 110 °C to prevent water vapor condensation on the sample that would have led to cracks and inhomogeneities. Then, the samples were heated up to 310 °C with a heating rate of 3 °C/min. Once the maximum

temperature was reached, it was maintained for 30 min. Afterwards, the samples were cooled down to room temperature with a cooling rate of 15 °C/min.

The growth processes were carried out through two different methods: conventional thermal annealing (CTA) [27] and flash heating (FH) [28–30]. The idea was to study the influence of the heating ramp in the pinning landscape originating within the films and, therefore, in the final superconducting properties. In the CTA method the samples were heated up to the corresponding $T_g$ (ranging from 730 to 810 °C), with a heating ramp of 25 °C/min. The atmosphere was humidified to decompose the $BaF_2$ precursor at high temperatures. Once the maximum temperature was reached, it was maintained for 180 min, and afterwards the samples were cooled down to 600 °C with a cooling rate of 2.5 °C/min. An oxygenation process at 600 °C was performed immediately after the growth. In the FH method the samples were immediately subjected to the corresponding $T_g$ (ranging from 700 to 810 °C) with a very fast heating ramp (1200 °C/min). The subsequent steps were performed identically as in the CTA method.

The surface morphology of the YBCO films was investigated using a scanning electron microscope (SEM, FEI Quanta 200 FEG). The phase and texture analyses of the YBCO thin films were performed using a two-dimensional 2D-XRD Bruker AXS General Area Detector Diffraction System (GADDS) diffractometer operating with Cu K$\alpha$. The transition temperature ($T_c$), the temperature at which the superconducting transition occurs, and the critical current density ($J_c$), the maximum current that can be applied without dissipation, were obtained inductively from low field zero field cooling (ZFC) and hysteretic magnetization measurements, respectively, performed with a superconducting quantum interference device (SQUID) magnetometer (Quantum Design, San Diego, CA, USA). The $J_c$ values were obtained using the Bean critical state model approach [31,32].

## 3. Results and Discussion

### 3.1. Morphological Characterization

SEM characterization was applied to determine the surface morphology of the grown YBCO layers. In Figures 1 and 2, SEM images of YBCO layers grown at different temperatures using CTA and FH methods are shown. As a general tendency, it can be observed that the higher the temperature, the lower the porosity. It should be noted, however, that several holes can be observed on the CTA sample grown at 810 °C (Figure 2f), generated through a temperature-induced dewetting phenomenon [33]. Regarding the samples grown with the CTA method, no *ab* grains are present in the working temperature range (730–810 °C). Concerning the samples grown with the FH annealing, some *ab* grains (identified as elongated needles at 90° in the SEM images) are detected in the temperature range 700–730 °C, and their density decreases with increasing temperature. No *ab* grains are observed in any of the films for the samples grown. Thus, SEM images suggest that the minimum temperature to assure *c*-axis nucleation for both CTA and FH is 750 °C.

The crystallinity of the samples was studied through X-ray diffraction (XRD). Figure 3 shows as an example the 2D-XRD frames obtained for an epitaxial CTA sample grown at 750 °C and an FH sample grown at very low temperature (700 °C) where non-fully epitaxial YBCO was obtained. In the first case, a highly epitaxial, i.e., *c*-axis oriented, YBCO film was attained, with the YBCO (00l) peaks (YBCO (003), YBCO (004), YBCO (005) and YBCO (006)) clearly identified. One can also see a spot with less intensity that corresponds to the K$_\beta$ peak of LAO. In the case of the FH sample grown at 700 °C, shown as an example of a non-fully-epitaxial sample, one can clearly identify the polycrystallinity of the YBCO film with two spots associated with YBCO (102) (*ab* grains) and a ring associated with YBCO (103) (random orientation). In order to better complete the XRD analysis, the integrated 2D-XRD patterns of samples grown at different temperatures through both the CTA and FH methods are shown in Figure 4. Regarding CTA samples (left column), we can see that at all four growth temperatures evaluated (730, 750, 770 and 810 °C) we obtain epitaxial films with *c*-axis YBCO (00l) peaks. It should be noted that, regardless the growth temperature used, a very small contribution of $Y_2Cu_2O_5$ (with (211) and (204)

reflections) can be identified, which is consistent with the barium deficiency imposed by the stoichiometry of the solutions.

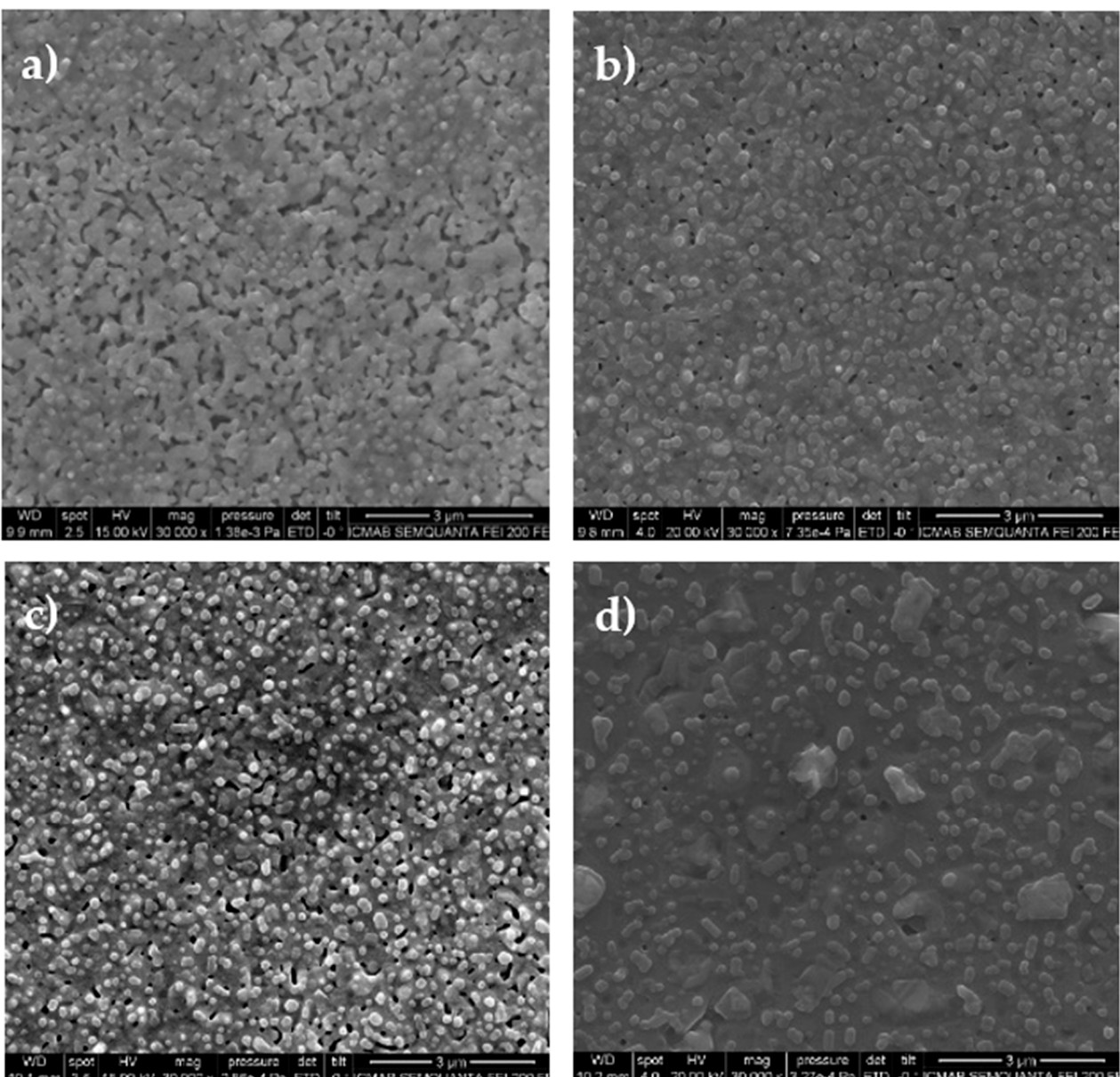

**Figure 1.** SEM images of different samples grown with the conventional thermal annealing (CTA) method at: (**a**) 730, (**b**) 750, (**c**) 770 and (**d**) 810 °C.

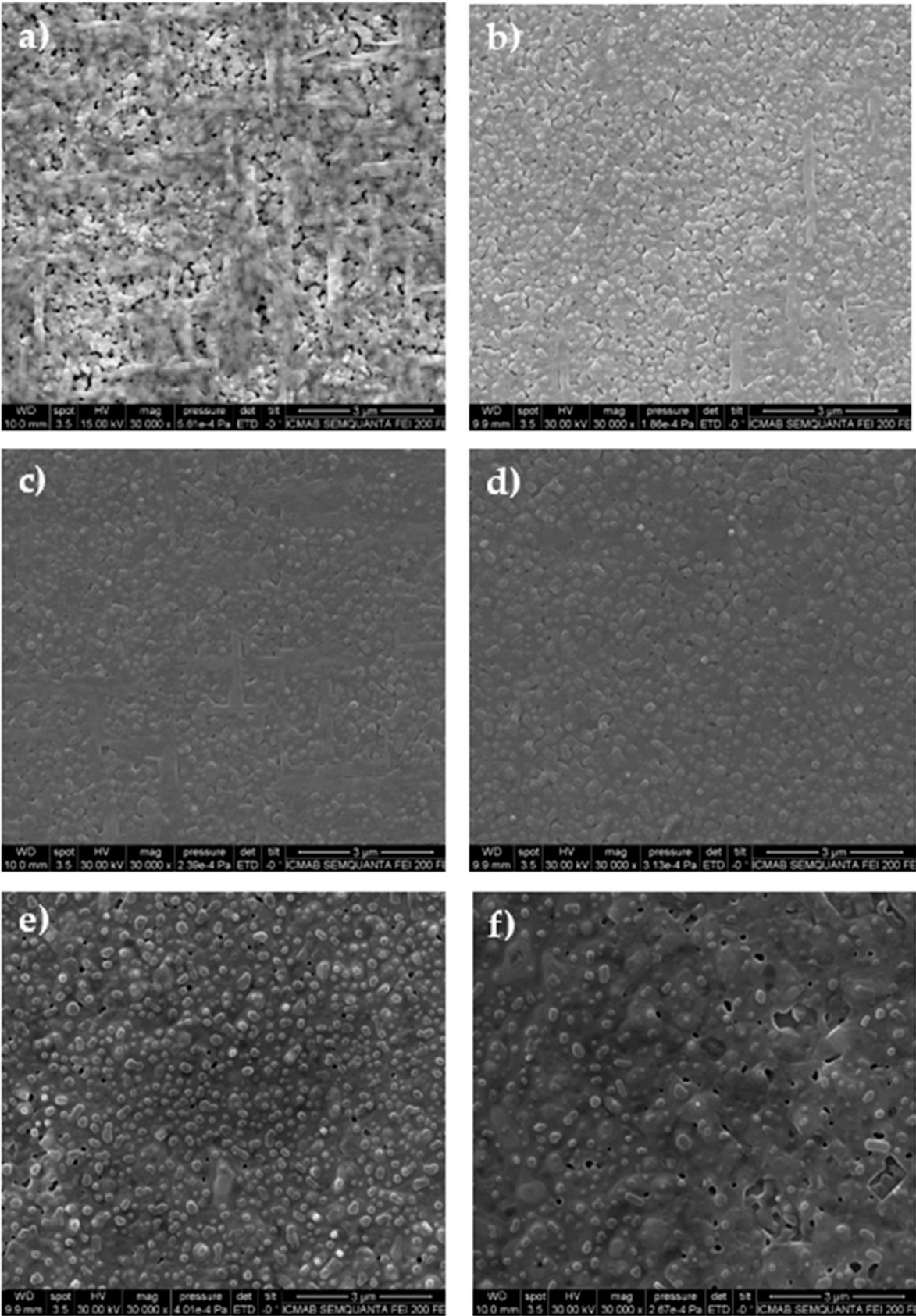

**Figure 2.** SEM images of different samples grown with the flash heating (FH) method at: (**a**) 700, (**b**) 720, (**c**) 730, (**d**) 750, (**e**) 780 and (**f**) 810 °C.

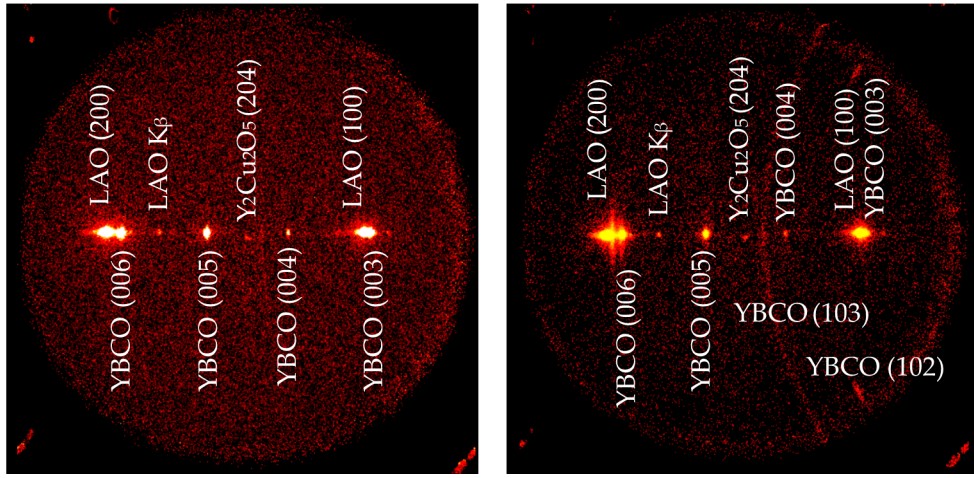

**Figure 3.** Two-dimensional XRD frames ($\chi$ vs. $2\theta$) from a sample grown at 750 °C with the CTA method (**left**) and 700 °C with the FH method (**right**).

**Figure 4.** Integrated X-ray diffraction patterns obtained from eight YBCO films grown at different temperatures. The (**left**) panels correspond to the samples grown with the CTA method and the (**right**) ones to the samples grown with the FH method.

Regarding FH samples (right column), we can see that at 730 and 750 °C we observe the same epitaxial YBCO (00l) peaks as those obtained for the CTA samples and also a small contribution of $Y_2Cu_2O_5$ (211). It is worth noting that for the FH samples grown at lower temperatures (700 and 720 °C), apart from YBCO epitaxial peaks we can also observe different polycrystalline reflections $Y_2Cu_2O_5$ (211), YBCO (103) CuO (11-1) and $Y_2O_3$ (113), where the YBCO (103) peak indicates a certain percentage of randomly oriented grains (see also Figure 3) in the films. Figure 5 shows the (103) YBCO phi-scan (φ-scan) and (005) YBCO rocking curve (ω-scan) obtained for CTA and FH samples grown at 750 °C, which reveal the in-plane and out-of-plane texture quality of both films. It is worth noting, therefore, that films with a fairly good texture quality and a high purity are obtained at temperatures as low as 750 °C when Ba-deficient solutions are used. This result is reminiscent of what has been previously observed in YBCO films including a certain amount of Ag in their composition where an extended growth window was identified [8,34]. The main reason for being able to grow *c*-axis oriented YBCO films at lower temperatures in the TFA process is associated with a modified nucleation behavior, very likely due to a shift of the stability line of YBCO with Ba-poor compositions towards a lower temperature, which then also shifts the crossover temperatures from *c*-axis to ab-axis nucleation [35,36].

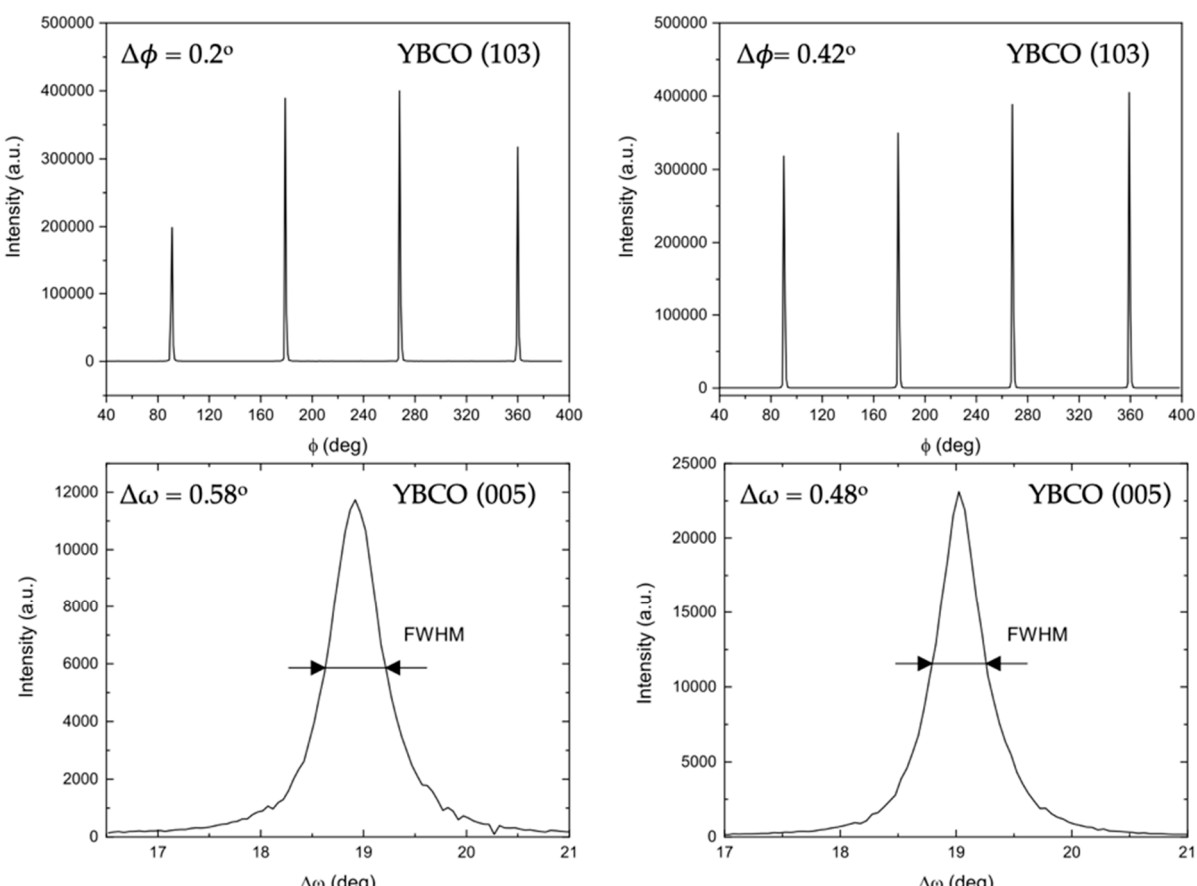

**Figure 5.** φ-scan (103) YBCO and ω-scan (005) YBCO obtained for YBCO films grown at 750 °C with the CTA (**left**) and FH (**right**) methods.

### 3.2. Superconducting Performance

The self-field critical current density, i.e., $J_c$ measured at zero external field ($J_c{}^{sf}$), was evaluated for several samples grown at different temperatures in order to compare their superconducting properties. $J_c{}^{sf}$ values obtained as a function of $T_g$ at 5 and 77 K for both CTA and FH methods are represented in Figure 6, where certain tendencies can be

identified. First, $J_c^{sf}$ increases with $T_g$ until the growth temperature reaches a value of 750 °C, where optimal nucleation and epitaxial growth are already achieved (trend shown with dashed red lines). For $T_g$ above 750 °C, the $J_c^{sf}$ values show an opposite trend (dotted blue lines). In this region, the increase of $J_c^{sf}$ with decreasing growing temperature is associated with an enhancement of pinning defects as shown in the following. Therefore, competition between good percolation and pinning performance gives us an optimal $T_g$ of 750 °C, where very high critical current values were obtained, with maximum values of $J_c^{sf}$ (77 K) = 6.6 and 6.9 MA/cm$^2$ for CTA and FH samples, respectively. The behavior shown in the superconducting performance of Ba-deficient samples grown at different temperatures is clearly different to the one observed on conventional stoichiometric YBCO films. In the former a $J_c^{sf}$ peak is observed, while in the latter $J_c^{sf}$ increases until a plateau of optimal epitaxial growth is reached [34]. It is important to note, moreover, that in order to obtain high $J_c^{sf}$ values in the range of 5 MA/cm$^2$ with standard (non-Ba-deficient) precursor solutions, a much higher $T_g$ (810 °C) is needed [8,37]. We can thus conclude that Ba-deficiency solutions enable to grow very high critical current density films at reduced temperatures of 750 °C on LAO substrates. This seems to indicate that the percolating currents along the low angle grain boundary network of the YBCO films are optimized at lower temperatures in Ba-deficient YBCO films [22].

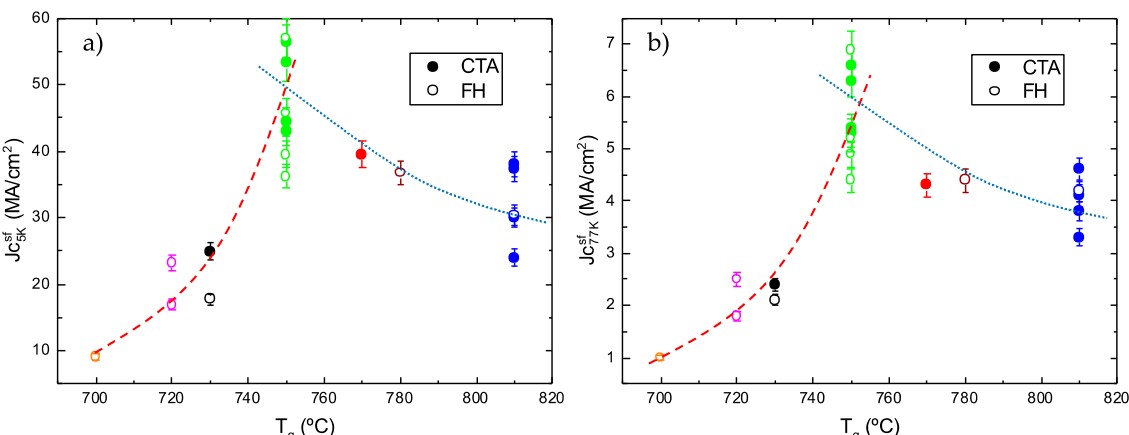

**Figure 6.** Self-field $J_c$ as a function of the growth temperature for samples grown by CTA (closed symbols) and FH (open symbols) at (**a**) 5 K and (**b**) 77 K. The wider scattering of data points at 750 °C and 810 °C comes from the fact that a large number of samples were tested at those temperatures. Dashed and dotted lines are provided as a guide.

The magnetic field dependence of $J_c$ at 5 K for samples grown at different temperatures is plotted in Figure 7. It is well-known that the log–log plot of the $J_c(H)$ dependence of YBCO thin films shows two different regions [38]. At low fields, where the number of vortices is less or equal to the number of defects, there is a plateau of $J_c$ associated with the single vortex pinning regime. At intermediate fields, where the number of vortices is equal or greater than the number of defects, the vortex–vortex interactions become relevant, resulting in a decrease of $J_c$, and a collective vortex motion is governed by a power law regime. The transition from single vortex to collective pinning regimes is defined by the crossover field $H^*$, which in our case was determined at 90% of self-field $J_c$.

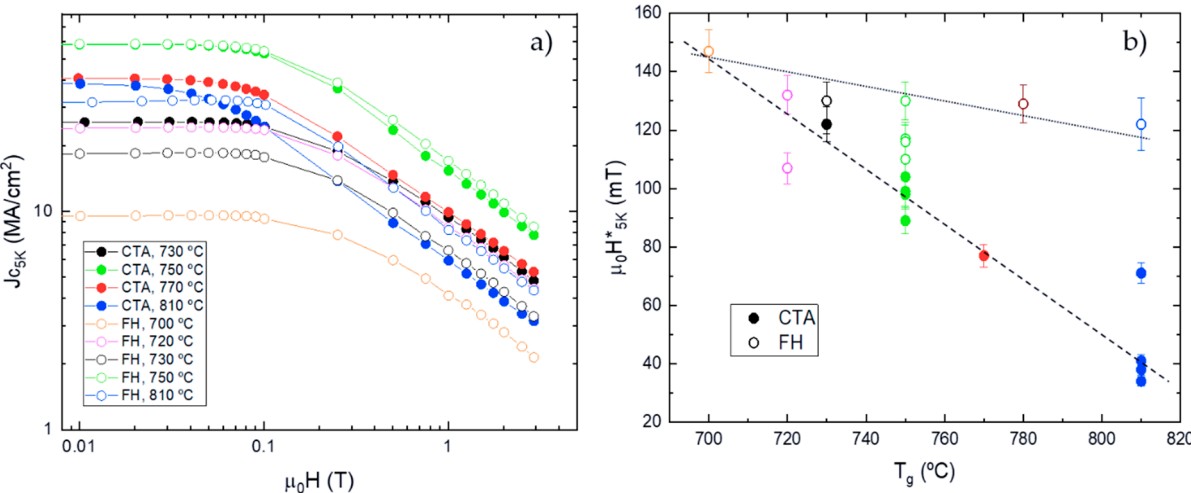

**Figure 7.** $J_c$ behavior with the magnetic field for samples grown with the CTA method (closed symbols) and the FH method (open symbols): (**a**) $J_c$ as a function of the magnetic field at different growth temperatures, (**b**) $\mu_0 H^*$ values determined at 5 K as a function $T_g$.

Figure 7b shows the $\mu_0 H^*$ values obtained for the different films as a function of $T_g$. Samples grown by the CTA method show a clear trend with a continuous enhancement of $\mu_0 H^*$ by reducing the growth temperature, indicating that more pinning centers are induced as we decrease $T_g$. In the case of FH films, although a slight enhancement of $\mu_0 H^*$ is observed at low $T_g$, a clear tendency cannot be identified. For those samples, the value of $\mu_0 H^*$ is already very high at $T_g$ = 810 °C, thus pointing out that the higher nucleation rate of FH process also helps to induce pinning defects within the matrix, which are not present with CTA growth [30]. The effect of rapid annealing on the defect structure and final properties of different functional materials has also been reported [39,40].

Figure 8a shows the temperature dependence of $J_c^{sf}$ for CTA and FH YBCO layers grown at different $T_g$. The $J_c(T)$ curves have been used to evaluate the contribution of strong and weak pinning centers, by fitting their expected $J_c(T)$ dependences [41]. Strong pinning shows a smooth decay with temperature (Equation (1)) [42], while weak pinning is characterized by a fast $J_c$ decay with temperature (Equation (2)) [43].

$$J_c^{weak}(T) \sim J_{c0}^{wk} e^{-T/T_0} \tag{1}$$

$$J_c^{strong}(T) \sim J_{c0}^{str} e^{-3\left(\frac{T}{T^*}\right)^2} \tag{2}$$

where $J_{c0}^{wk}$ and $J_{c0}^{str}$ are the contributions to $J_c$ at 0 K of weak and strong pinning defects, respectively, and $T_0$ and $T^*$ their characteristic pinning energies. Figure 8b shows the contribution of $J_c$ associated with strong pinning defects at 0 K, obtained by fitting Equations (1) and (2) in the curves depicted in Figure 8a. We observe that samples grown at 810 °C show a similar weight of strong and weak contributions (~50–60% of strong pinning). However, when reducing the growing temperature, the contribution of strong pinning increases, completely dominating the pinning landscape in samples grown at 700 °C (90% of strong pinning). These results indicate that, as previously observed in the $J_c(H)$ dependence, the growing temperature strongly affects the pinning landscape generated within the YBCO matrix. Further analysis of angular $J_c$ dependences in conjunction with an advanced TEM microstructural analysis should be performed in order to elucidate the nature of pinning centers. However, considering the results obtained in other CSD films and nanocomposites studied, an enhancement of $\mu_0 H^*$ and isotropic strong pinning contribution can be correlated with the presence of stacking faults and associated nanostrain [29,30,44,45].

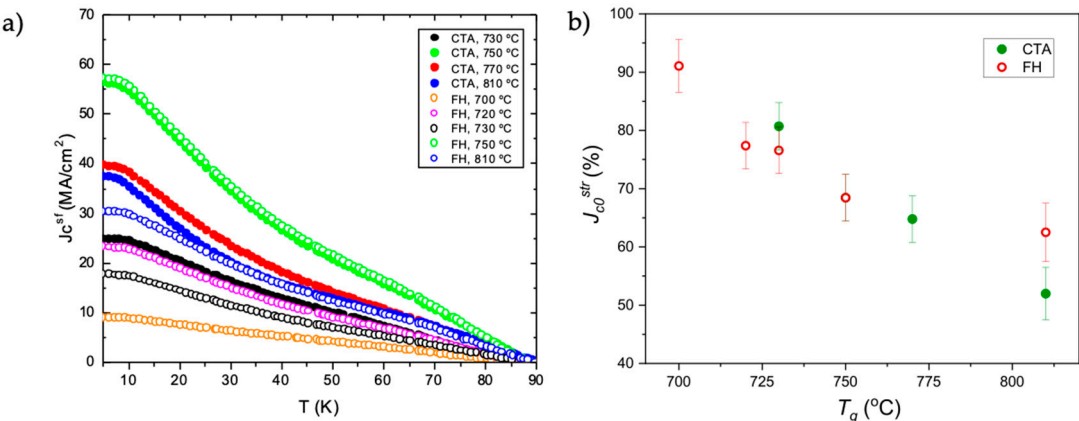

**Figure 8.** (**a**) Self-field critical current density $J_c$ values as a function of the temperature for samples grown with the CTA method (closed symbols) and the FH method (open symbols) (**b**) Percentage of strong pinning contribution to $J_c$ at 0 K, determined by fitting the $J_c(T)$ curves shown in (**a**) with Equations (1) and (2).

## 4. Conclusions

In this work we evaluated the effect of the growing temperature on the superconducting performance of YBCO films grown by Ba-deficient low-fluorine solutions using two different approaches, conventional thermal annealing (CTA) and flash heating (FH). The obtained results demonstrate that in both cases, substantial improvements can be obtained by using Ba-deficient solutions, which allow us to reduce the growth temperature of YBCO films. Samples grown at lower $T_g$ show smoother $J_c(H)$ dependencies, with higher $\mu_0 H^*$ values, and strong pinning contribution, which indicate that a higher density of defects are achieved in the YBCO matrix. A good compromise between epitaxial growth, with optimized current percolation, and pinning performance gives us an optimum $T_g$ = 750 °C, where high self-field $J_c$ values are obtained (7 MA/cm$^2$ at 77 K). The use of lower growth temperatures, as compared with stoichiometric YBCO films, may be also helpful when growing on top of technical substrates, preventing chemical reaction with substrate buffer layers.

**Author Contributions:** P.T., J.A., L.P., C.P. grew the samples and performed the physical and morphological analysis. S.R., N.M., X.O., T.P., G.S., G.C., and A.P. provided supervision. P.T., J.A., A.P. wrote the manuscript. All authors have read and agreed to the published version of the manuscript.

**Funding:** This work has been carried out within the framework of the EUROfusion Consortium and has received funding from the Euratom Research and Training Programme 2014–2018 and 2019–2020 under grant agreement No 633053. The views and opinions expressed herein do not necessarily reflect those of the European Commission. We acknowledge financial support from Spanish Ministry of Economy and Competitiveness through the Severo Ochoa Programme for Centres of Excellence in R&D (SEV-2015-0496, FUNFUTURE CEX2019-000917-S), SUMATE project RTI2018-095853-B-C21, cofinanced by the European Regional Development Fund, COST-Nanocohybri CA16218 and Catalan Government with 2017-SGR-1519 and XRE4S.

**Data Availability Statement:** No new data were created or analyzed in this study. Data sharing is not applicable to this article.

**Acknowledgments:** J.A. would like to thank the UAB PhD program in Materials Science.

**Conflicts of Interest:** The authors declare no conflict of interest.

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
