# Peer review of "Low-Fluorine Ba-Deficient Solutions for High-Performance Superconducting YBCO Films"

_coatings, doi:10.3390/coatings11020199_

Round 1
Reviewer 1 Report
Dear Editor,
I believe that figure 2b, 2c, and 2d are not the original one. They blurred and defocused. One can notice it by how also the text at the bottom is blurred.
Kind regards, stay safe
Author Response
We thank the referee for drawing our attention to the quality of figure 2. It has been corrected in the new version.
Reviewer 2 Report
This manuscript is interesting and provides the original data and findings.
Some items of comments are as follows:
- As mentioned in section 3.1, “As a general tendency, it can be observed that the higher the temperature, the lower the porosity.” It seems that the size of pore suddenly becomes larger at temperature 810°C for samples grown with FH method, shown in Fig. 2f). Why?
- Self-field Jc at an optimal Tg of 750°C, as shown in Fig. 5, exhibits a wide scattering from 35-60 MA/cm2 at 5 K and 4-7 MA/cm2at 77 K, in comparison to other cases at different Tg. Similar situation occurs for the cases at Tg = 810°C, Why?
- Physical meaning and significance of “self-field Jc” and other important properties had better to be briefly described in the text for the convenience of readers.
- There is no corresponding bracket “(“ in the text “…less than 0.6%).” in section 2, line.73.
- In comparison with the superconducting YBCO films grown by other currently popular methods, the discrepancy in properties and performance of these films should be briefly described and compared in the text.
Reviewer 3 Report
In this manuscript, the authors demonstrate the effect of the growth temperature on the superconducting performance of YBCO films grown by Ba-deficient low-fluorine solutions using two different approaches, the conventional thermal annealing (CTA) and flash heating (FH) methods. The obtained results demonstrate that a lower growth temperature can be realized by using Ba-deficient solutions, which results in better film quality and superconductive performance. The work in this manuscript systematically studied the influence of the growth temperature from crystal morphology, orientation, and device performance. Some suggestions are listed.
- The authors may think about more approaches to prove the proposed “epitaxial growth.” From the SEM image, it seems the orientation of the crystals is random. Even though 2D-XRD gives the preferred orientations, more evidence is suggested since the claim of “epitaxial growth” normally requires lattice-level characterizations.
- In Figure 4, the omega scan may provide more details on the crystal quality change with different growth temperatures.
- In Figure 6b, the proposed tendency with the data points is not very convincing.
Reviewer 4 Report
Review attached.

Author Response
Reviewer 4
- The authors propose that methanol was employed as a solvent in the preparation of the YBCO sol. It is well reported in literature that the properties of the sol and the films is a function of the organic solvent. Can the authors comment on this aspect and justify their choice of using methanol as solvent as opposed to ethanol, which is widely used solvent in sol-gel depositions?
We have added a reference [Obradors et al. SUST 25, 123001 (2012)] were the influence of the solvent on the properties of the films is well reported. The choice of methanol was performed according to the best results obtained on those studies.
- It is interesting to know if the authors performed a time-dependent flash annealing study. Can the authors comment on the crystallization/defects of the YBCO films in the flash annealing process when the annealing time is varied from 180 min to 30 min as there are reports that show rapid crystallization of films after 5 min of annealing.
This is a very good point that deserve a detailed study. We have not studied here the dependence of crystallization/defects by changing the annealing time since the paper was mainly focused on studding the effect of the growing temperatures. We think however, that the proposed study is very interesting and thus will be performed in further investigations of Ba-deficient films.
- There are several published articles on rapid annealing such as “Bhosle, S. M.; Friedrich, C. R., Rapid Heat Treatment for Anatase Conversion of Titania Nanotube Orthopedic Surfaces. Nanotechnology 2017, 28, 405603” and “Darapaneni, P.; Moura, N. S.; Harry, D.; Cullen, D. A.; Dooley, K. M.; Dorman, J. A., Effect of Moisture on Dopant Segregation in Solid Hosts. The Journal of Physical Chemistry C 2019, 123, 12234-12241” that discuss on the defects and their influence on the properties of the nanostructures. It is recommended to cite such relevant literature in this article.
This works have been cited and mentioned in the text
Round 2
Reviewer 1 Report
Dear Authors and Reviewer #2,
the present manuscript is highly improved versus the original version, and I please to agree to accept as it is.
Stay safe, kind regards